# STK25 suppresses Hippo signaling by regulating SAV1-STRIPAK antagonism

**Sung Jun Bae[1], Lisheng Ni[1], Xuelian Luo[1,2]***

[1]Department of Pharmacology, University of Texas Southwestern Medical Center, Dallas, United States; [2]Department of Biophysics University of Texas Southwestern Medical Center, Dallas, United States

**Abstract** The MST-LATS kinase cascade is central to the Hippo pathway that controls tissue homeostasis, development, and organ size. The PP2A complex STRIPAK[SLMAP] blocks MST1/2 activation. The GCKIII family kinases associate with STRIPAK, but the functions of these phosphatase-associated kinases remain elusive. We previously showed that the scaffolding protein SAV1 promotes Hippo signaling by counteracting STRIPAK (Bae et al., 2017). Here, we show that the GCKIII kinase STK25 promotes STRIPAK-mediated inhibition of MST2 in human cells. Depletion of STK25 enhances MST2 activation without affecting the integrity of STRIPAK[SLMAP]. STK25 directly phosphorylates SAV1 and diminishes the ability of SAV1 to inhibit STRIPAK. Thus, STK25 as the kinase component of STRIPAK can inhibit the function of the STRIPAK inhibitor SAV1. This mutual antagonism between STRIPAK and SAV1 controls the initiation of Hippo signaling.

## Introduction

The Hippo pathway is critical for organ size control, tissue homeostasis, development, and tumor suppression by restricting cell proliferation and promoting apoptosis (*Harvey and Tapon, 2007*; *Pan, 2010*; *Johnson and Halder, 2014*; *Yu et al., 2015*; *Fu et al., 2017*; *Misra and Irvine, 2018*). Dysregulation of the Hippo pathway is linked to tumor initiation and progression in flies, mice, and humans (*Pan, 2010*; *Halder and Johnson, 2011*; *Yu and Guan, 2013*; *Yu et al., 2015*). In humans, the core kinase cascade of the Hippo pathway consists of Sterile-20 family kinases MST1/2, NDR family kinases LATS1/2, the scaffolding protein SAV1, and the adaptor protein MOB1. When the Hippo pathway is on, the MST1/2–SAV1 complexes phosphorylate and activate the LATS1/2–MOB1 complexes (*Chan et al., 2005*; *Praskova et al., 2008*). Activated LATS1/2–MOB1 then phosphory-late transcriptional coactivators YAP/TAZ. Phosphorylation of YAP/TAZ results in their cytoplasmic sequestration and degradation (*Dong et al., 2007*; *Zhao et al., 2007*; *Zhao et al., 2010*). When the Hippo pathway is off, YAP/TAZ localize to the nucleus and form functional hybrid transcription factors with sequence-specific DNA-binding proteins TEAD1-4. The YAP/TAZ–TEAD transcription factors induce the expression of Hippo-target genes to enable cell proliferation and survival (*Zhao et al., 2008*; *Luo, 2010*).

Activation of MST1/2 kinases initiates Hippo signaling (*Zhou et al., 2009a*; *Song et al., 2010*). MST1/2 contain an N-terminal kinase domain, a C-terminal SARAH dimerization domain, and a flexi-ble linker between the two. MST1 and MST2 can each form a constitutive homodimer through the SARAH domain, and their activation requires trans-autophosphorylation of the activation loop (T183 for MST1 and T180 for MST2) (*Avruch et al., 2012*; *Jin et al., 2012*; *Ni et al., 2013*). Activated MST1/2 autophosphorylate multiple threonine residues in the linker region (*Ni et al., 2015*). The forkhead-associated domain (FHA) of the sarcolemmal membrane-associated protein (SLMAP), an adaptor protein of the multi-subunit PP2A complex STRIPAK (striatin-interacting phosphatase and kinase), recognizes the phospho-threonine residues in the MST1/2 linker region. Binding of SLMAP to the phospho-MST2 linker recruits STRIPAK and promotes PP2A-mediated dephosphorylation of

**\*For correspondence:**
xuelian.luo@utsouthwestern.edu

**Competing interests:** The authors declare that no competing interests exist.

pT180, thus ensuring low steady-state MST2 activation in fast dividing cells (*Bae et al., 2017*; *Zheng et al., 2017*). Thus, STRIPAK inhibits MST1/2 activation in a feedback mechanism. The MST1/2-associated protein SAV1 promotes MST2 activation by suppressing STRIPAK$^{SLMAP}$-mediated dephosphorylation of MST2 T-loop (*Bae et al., 2017*). The N-terminal region of SAV1 contains a phosphatase-inhibitory domain (PID) that directly associates with PP2A and inhibits its phosphatase activity in vitro. Thus, SAV1 activates MST1/2 by inhibiting the inhibitor STRIPAK$^{SLMAP}$. These findings suggest dynamic antagonism between SAV1 and STRIPAK$^{SLMAP}$.

As a major serine/threonine phosphatase, PP2A consists of the scaffold subunit A, the catalytic subunit C, and the regulatory subunit B (*Shi, 2009*). STRIPAK is an unusually large PP2A assembly (*Hwang and Pallas, 2014*; *Shi et al., 2016*). In addition to the A/C catalytic core and the PP2A regulatory B‴ subunit striatin (STRN3 in humans), STRIPAK also contains STRIP1, MOB4, SIKE1, CCM3, and one GCKIII kinase (MST3, MST4, or STK25) (*Goudreault et al., 2009*; *Ribeiro et al., 2010*; *Couzens et al., 2013*). These STRIPAK components form a complicated protein-protein interaction network, with STRN3 serving as a major scaffold (*Tang et al., 2019*). STRN3 comprises an N-terminal coiled-coil (CC) domain, a middle region, and a C-terminal WD40-repeat domain. STRN3 CC directly binds to PP2AA/C. The WD40-repeat domain interacts with MOB4. The middle region directly associates with CCM3, which in turn recruits GCKIII kinases to STRN3. STRN3 also directly interacts with STRIP1 and SIKE1. Recent studies have established STRIPAK as a platform to integrate upstream inputs to regulate the Hippo pathway (*Bae et al., 2017*; *Chen et al., 2019*).

The GCKIII kinase family is involved in various cellular processes, including cell proliferation, transformation, migration, polarity, apoptosis, Golgi assembly, and cell cycle progression (*Sugden et al., 2013*). It has three members: STK25 (Serine/Threonine kinase 25; also called SOK1 or YSK1), MST3 (mSte20-like kinase3, also known as STK24), and MST4 (also known as STK26 or MASK). GCKIII kinases contain a conserved N-terminal kinase domain, a C-terminal dimerization domain, and a linker between the two. Depletion of *Drosophila* GCKIII or Cka (a homolog of human STRNs) has similar effects on suppressing ectopic wing veins (*Friedman and Perrimon, 2006*; *Horn et al., 2011*), suggesting that GCKIII kinases may promote STRIPAK function and suppress the Hippo pathway. On the other hand, it has been recently reported that STK25 promotes Hippo pathway activation through directly activating LATS1/2 (*Lim et al., 2019*). Therefore, the roles of GCKIII kinases in the Hippo pathway remain unclear.

In this study, we clarify the functions of GCKIII kinases during Hippo signaling. We show that among the three GCKIII kinases, only STK25 regulates MST1/2. Similar to other STRIPAK components, STK25 suppresses Hippo pathway activation. One mechanism by which it does so is to phosphorylate SAV1 and antagonize the ability of SAV1 to inhibit PP2A. Thus, our study further extends the intricate, dynamic antagonism between STRIPAK and SAV1, and demonstrates the importance of the delicate balance between kinases and phosphatases in Hippo activation.

## Results

### STK25 inhibits the Hippo pathway in human cells

We individually depleted each GCKIII kinase from 293FT cells by RNA interference (RNAi) and monitored MST2 activation by examining the levels of MST2 T180 phosphorylation (pT180). Among the three GCKIII kinases, only depletion of STK25, but not depletion of MST3 or MST4, increased MST2 pT180 (*Figure 1—figure supplement 1A*). Conversely, overexpression of STK25, but not overexpression of MST3 or MST4, decreased MST2 pT180 (*Figure 1—figure supplement 1B*). These results suggest that, among the three GCKIII kinases, only STK25 is involved in suppressing MST2 activation.

We next deleted each GCKIII kinase from 293A cells with CRISPR (Clustered regularly interspaced short palindromic repeats)/Cas9. Compared to control cells, only STK25 knockout (KO) cells, but not MST3 KO or MST4 KO cells, showed increased T-loop phosphorylation of MST1/2 (pMST1/2) and elevated MOB1 phosphorylation at T35 (*Figure 1A* and *Figure 1—figure supplement 1C*). Even in the absence of contact inhibition, phosphorylation of YAP was increased in STK25 KO cells. Consistent with the spontaneous activation of the Hippo pathway, a higher percentage of STK25 KO cells,

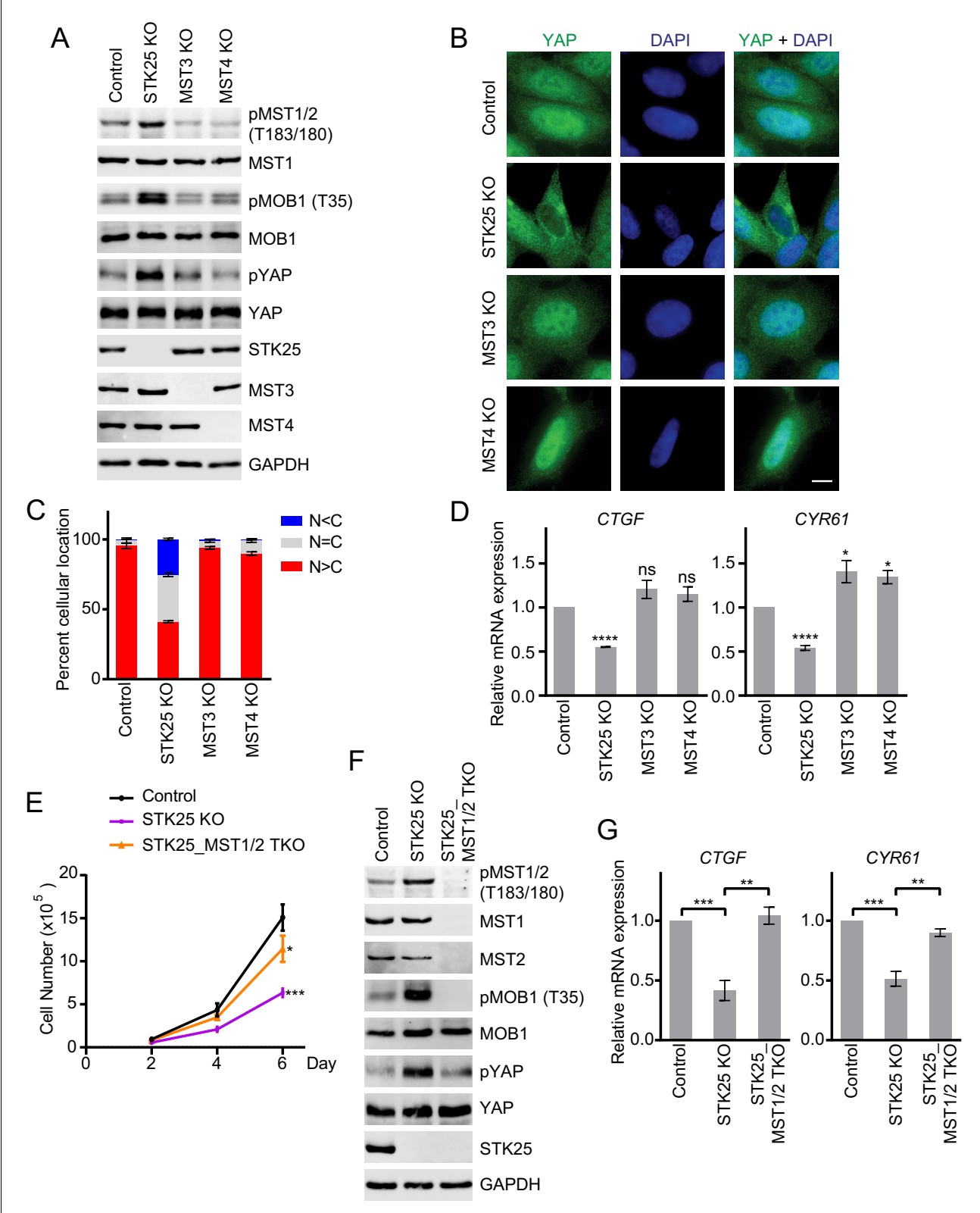

**Figure 1.** STK25 inhibits the Hippo pathway in human cells. (**A**) Immunoblots of total lysates of control, and GCKIII (STK25, MST3 and MST4) knockout (KO) 293A cells with the indicated antibodies. Anti-GAPDH blot was used as the loading control. (**B**) Immunofluorescence staining of YAP localization in control and the indicated GCKIII KO 293A cells. Cells were fixed, permeabilized, and stained with anti-YAP (green) and DAPI (blue). Scale bar, 5 µm. (**C**) Quantification of immunofluorescence signal intensities in (**B**). Approximately 100 cells were counted from five random fields of each sample. N < C

*Figure 1 continued on next page*

*Figure 1 continued*

(blue), N = C (grey), and N > C (red) categories indicate YAP localization in cytoplasm, both cytoplasm and nucleus, and nucleus, respectively. Data are plotted as mean ± SEM of three biological replicates. (D) Quantitative RT-PCR. Relative mRNA expression of YAP target genes *CTGF* and *CYR61* in control and the indicated GCKIII kinase KO 293A cells. Data are plotted as mean ± SEM of three biological replicates (*p<0.05; ****p<0.0001; ns, non-significant). (E) Cell proliferation assay was performed in 293A cells. Cell proliferation curves in control (black), STK25 KO (purple), and STK25_MST1/2 TKO (orange) cells were plotted, respectively. Cells were counted on days 2, 4, and 6 after seeding. Data shown are the means ± SEM of three independent experiments. Numbers of STK25 KO or STK25_MST1/2 TKO cells on day 6 was compared to that of control cells (*p<0.05; ***p<0.001). (F) Immunoblots of control, STK25 KO, and STK25_MST1/2 TKO 293A cell lysates with the indicated antibodies. (G) Relative mRNA expression of YAP target genes *CTGF* and *CYR61* in control, STK25 KO, and STK25_MST1/2 TKO 293A cells. Data are plotted as mean ± SEM of three biological replicates (**p<0.01; ***p<0.001).

The online version of this article includes the following figure supplement(s) for figure 1:

**Figure supplement 1.** STK25 inhibits the Hippo pathway in human cells.
**Figure supplement 2.** STK25 inhibits MST2-mediated LATS1 activation.
**Figure supplement 3.** STK25 depletion does not inhibit Hippo activation induced by cytoskeleton disruption.
**Figure supplement 4.** STK25 depletion does not affect the F-actin cytoskeleton.

but not MST3 KO or MST4 KO cells, exhibited cytoplasmic localization of YAP (*Figure 1B and C*). The expression of two well-established Hippo target genes, *CTGF* and *CYR61*, was reduced in STK25 KO cells, but not in MST3 KO or MST4 KO cells (*Figure 1D*). These results establish STK25 as an important negative regulator of the Hippo pathway.

## STK25 inhibits Hippo signaling by suppressing MST1/2 activation

To test whether spontaneous MST1/2 activation was responsible for elevated Hippo signaling in STK25 KO cells, we created STK25_MST1/2 triple KO (TKO) 293A cells with *STK25* and both *MST1* and *MST2* deleted. Consistent with an activated Hippo pathway, STK25 KO cells displayed much slower cell proliferation compared to control cells (*Figure 1E*). In contrast, STK25_MST1/2 TKO cells displayed only slightly reduced cell proliferation rate, as compared to control cells. Moreover, phosphorylation of MOB1 at T35 and phosphorylation of YAP were diminished in STK25_MST1/2 TKO cells (*Figure 1F* and *Figure 1—figure supplement 1D*). Expression of Hippo target genes, *CTGF* and *CYR61*, was also restored in STK25_MST1/2 TKO cells (*Figure 1G*). Thus, deletion of MST1/2 bypasses the function of STK25 in Hippo signaling, indicating that the major function of STK25 in the Hippo pathway is to suppress MST1/2 activation.

Recently, the Ganem group has reported that STK25 promotes the activation of the Hippo pathway (*Lim et al., 2019*), and activates LATS1/2 by increasing the phosphorylation of the LATS1/2 activation loop (AL) without altering hydrophobic motif (HM) phosphorylation. These results are inconsistent with ours. To examine whether STK25 can promote LATS1 activation without phosphorylation of the hydrophobic motif, we co-expressed LATS1 wild-type (WT) or a kinase-dead mutant (D828N) with STK25 or MST2, respectively. As expected, co-expression of MST2 increased both HM and AL phosphorylations of LATS1 WT (*Figure 1—figure supplement 2A*). MST2 also increased HM phosphorylation of LATS1 D828N, but failed to increase AL phosphorylation of this mutant, consistent with the notion that AL phosphorylation of LATS1 is mediated by autophosphorylation. In contrast, co-expression of STK25 did not increase the phosphorylation of either HM or AL of LATS1 WT or D828N. Furthermore, expression of STK25 actually inhibited MST2-mediated activation of LATS1 in a dose-dependent manner (*Figure 1—figure supplement 2B*). Finally, we performed in vitro kinase assays with purified recombinant proteins. Addition of the phosphorylated HM peptide activated the AL phosphorylation of the LATS1 kinase domain that lacked the HM (LATS1C-ΔHM) (*Figure 1—figure supplement 2C*). Neither STK25 nor MST2 was able to directly phosphorylate the AL of LATS1.

The Ganem group has further shown that STK25 depletion inhibits Hippo pathway activation that is induced by cytoskeleton disruption (*Lim et al., 2019*). To test whether STK25 indeed has a role in this process, we used two different compounds, latrunculin B (LatB) and cytochalasin D (CTD), to disrupt the actin cytoskeleton in control or STK25 KO 293A cells. Both LatB and CTD activated the Hippo pathway in control and STK25 KO cells (*Figure 1—figure supplement 3A and B*). However, there were no significant differences in Hippo pathway activation between control and STK25 KO cells under these conditions. Collectively, our results do not support direct or indirect roles of STK25

in activating LATS1/2 and downstream events. Instead, STK25 restrains MST1/2 activation to promote cell proliferation.

We further tested whether STK25 depletion affected F-actin cytoskeleton, which plays a central role in regulating the Hippo pathway. We stained F-actin with phalloidin (*Figure 1—figure supplement 4A*) and showed that there were no changes in F-actin organization and cell shape between control and STK25 KO cells. To test whether STK25 depletion affected cell size, we measured the cell area of control and STK25 KO cells based on F-actin staining. As shown in *Figure 1—figure supplement 4B*, control and STK25 KO cells had similar cell size. Taken together, depletion of STK25 did not affect the overall organization of F-actin cytoskeleton in 293 cells. Our results are consistent with the previously reported finding that GCKIII kinases MST3 and MST4, but not STK25, are involved in cytoskeletal regulation (*Madsen et al., 2015*).

## STK25 inactivates MST1/2 as a component of STRIPAK[SLMAP]

The FHA domain of SLMAP recognizes the phosphorylated MST2 linker, and positions PP2A to dephosphorylate pT180 of MST2 (*Figure 2A*). We next tested whether STK25 suppresses MST1/2 activation as a component of the STRIPAK[SLMAP] complex. STK25 overexpression reduced pT180 of MST2 WT without altering pT336 in the linker (*Figure 2B*), but could not reduce pT180 of the MST2 7TA mutant, which eliminated major phosphorylation sites in the linker region (*Ni et al., 2015*). Furthermore, MST2 7TA no longer interacted with SLMAP or STK25 (*Figure 2C*). These results indicate that the MST2-SLMAP interaction is required for STK25-dependent inactivation of MST2.

SIKE1 and CCM3 link SLMAP and STK25 to STRN3, respectively (*Figure 2A*). Depletion of either SIKE1 or CCM3 impaired the association between SLMAP and STK25 (*Figure 2D*). Moreover, MST2 pT180 was elevated when either SIKE1 or CCM3 was depleted by RNAi (*Figure 2E*). These results suggest that STK25 likely regulates MST2 pT180 as a component of the STRIPAK[SLMAP] complex.

We next explored the mechanism by which STK25 regulates MST1/2. Overexpression of STK25 WT or its kinase-dead mutant D158N did not affect the interaction between SLMAP and MST1 (*Figure 2—figure supplement 1A*). Furthermore, either STK25 overexpression or deletion did not change the association of STRN3 with other STRIPAK components (*Figure 2F* and *Figure 2—figure supplement 1B*). These observations indicate that STK25 is not required for the integrity of the STRIPAK complex or its interaction with MST1/2.

## STK25 inhibits the ability of SAV1 to counteract STRIPAK

Because SAV1 activates MST1/2 by antagonizing STRIPAK (*Bae et al., 2017*), we hypothesized that STK25 might suppress MST1/2 activation by inhibiting SAV1. To test this hypothesis, we deleted SAV1 in STK25 KO cells using the CRISPR/Cas9 system. The STK25_SAV1 double KO (DKO) cells showed higher proliferation rates, decreased pMST1/2, pMOB1, and pYAP levels, as compared to STK25 KO cells (*Figure 3A and B*, and *Figure 3—figure supplement 1A*). Furthermore, co-expression of STK25, but not MST3 or MST4, reduced SAV1-induced pT180 of MST2 (*Figure 3C*). STK25 overexpression did not alter the MST2-SAV1 or SAV1-SAV1 interactions (*Figure 3—figure supplement 1B and C*). These results suggest that STK25 can inhibit the ability of SAV1 to activate MST2. The kinase-dead mutant of STK25, D158N, had no effect on pT180 of MST2 (*Figure 3D*), indicating that the kinase activity of STK25 is required for SAV1 inhibition.

We tested whether STK25 could phosphorylate SAV1. Expression of STK25 WT, but not its kinase-dead mutant D158N, clearly retarded the mobility of SAV1 on phos-tag gels and CIP (Calf Intestinal Phosphatase) treatment reversed SAV1 retardation (*Figure 3E*). Thus, SAV1 was phosphorylated by STK25 in cells. The N-terminal phosphatase-interacting domain (PID) of SAV1 directly binds to and inhibits PP2A (*Bae et al., 2017*). We tested whether SAV1 PID was phosphorylated by STK25 in vitro. We performed in vitro kinase assays using recombinant PID (SAV1[1-90]) as the substrate and STK25 as the kinase in the presence of cold ATP, and analyzed the treated PID with mass spectrometry. Several STK25 phosphorylation sites were identified in SAV1 PID, including T26. We then raised a phospho-specific antibody against pT26. The antibody selectively recognized SAV1 PID that had been phosphorylated by STK25, but not SAV1 T26A (*Figure 3—figure supplement 1D*). Using this antibody, we confirmed that SAV1 WT was indeed phosphorylated by STK25 at T26 in human cells (*Figure 3F*). Phosphorylation of SAV1 T26 was diminished in STK25 KO cells, as compared to control cells (*Figure 3G*).

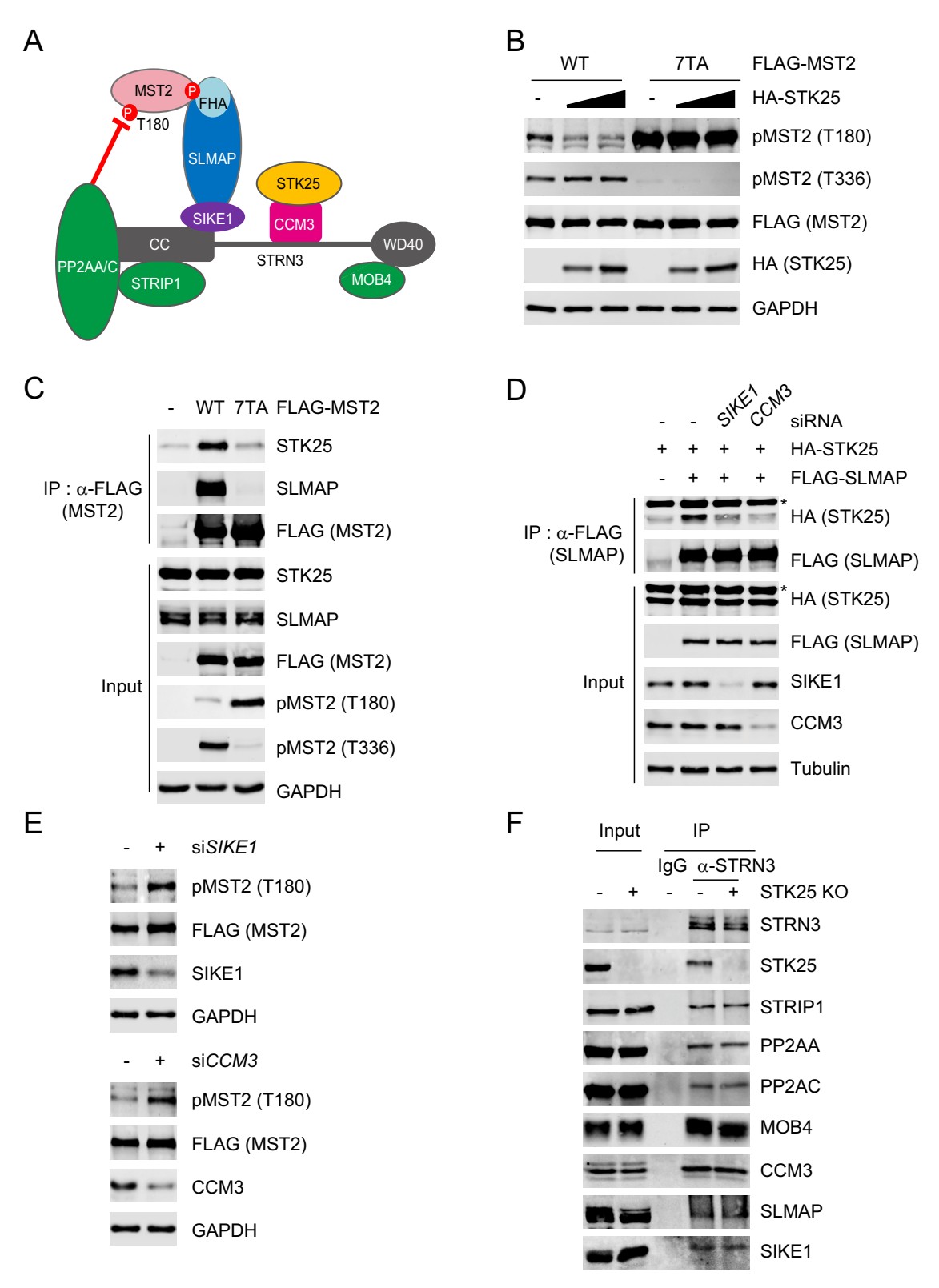

**Figure 2.** STK25 inactivates MST2 as a component of STRIPAK^SLMAP. (**A**) Model for STRIPAK^SLMAP-mediated MST2 inactivation. (**B**) 293FT cells were co-transfected with HA-STK25 and the indicated FLAG-MST2 plasmids. The total cell lysates were blotted with the indicated antibodies. (**C**) 293FT cells were transfected with the indicated FLAG-MST2 plasmids. The total cell lysates (input) and anti-FLAG immunoprecipitation (IP) were blotted with the indicated antibodies. Anti-GAPDH blot was used as the loading control. (**D**) 293FT cells were co-transfected with the indicated siRNA and plasmids.
*Figure 2 continued on next page*

Figure 2 continued

The total cell lysates (input) and anti-FLAG IP were blotted with the indicated antibodies. Anti-Tubulin blot was used as the loading control. Asterisk denotes a non-specific band. (E) Immunoblots of 293FT cell lysates co-transfected with the indicated siRNAs and FLAG-MST2. (F) The total cell lysates (input) and anti-IgG or anti-STRN3 IP of control 293A cells and STK25 KO 293A cells were blotted with the indicated antibodies.

The online version of this article includes the following figure supplement(s) for figure 2:

**Figure supplement 1.** STK25 is not required for STRIPAK assembly or its interaction with MST1/2.

We tested whether phosphorylation of SAV1 PID by STK25 prevented its binding to PP2A. Non-phosphorylated SAV1 PID formed a complex with PP2AA during gel filtration chromatography (*Figure 3H* and *Figure 3—figure supplement 1E*). In contrast, SAV1 PID phosphorylated by STK25 (pSAV1$^{1-90}$) did not interact with PP2AA and did not alter its fractionation behavior. Thus, STK25 phosphorylates SAV1 PID and blocks its binding to PP2A.

We next investigated if SAV1 could co-localize with STK25 in cells. STK25 has been previously shown to be targeted to and activated at the Golgi apparatus by the Golgi matrix protein GM130 (*Preisinger et al., 2004*). A major pool of STK25 indeed co-localized with GM130 at the Golgi (*Figure 3—figure supplement 2A*). Intriguingly, a pool of SAV1 also co-localized with STK25 at the Golgi and in the peri-nuclear region (*Figure 3—figure supplement 2B and C*). Thus, specific cellular regions, such as the Golgi apparatus and the peri-nuclear region, might organize and coordinate SAV1 phosphorylation by STK25 with Hippo signaling events.

## SAV1 phosphorylation by STK25 inhibits SAV1-dependent activation of MST1/2

We examined whether SAV1 phosphorylation by STK25 is functionally important for MST1/2 activation in human cells. Expression of SAV1 WT or T26A elevated pT180 of MST2, and STK25 co-expression still inhibited the ability of SAV1 T26A to induce MST2 activation (*Figure 4—figure supplement 1A*). To test if additional phosphorylation sites of SAV1 were also involved in Hippo activation, we mutated two other conserved phosphorylation sites (S36A and S68A) identified by mass spectrometry. Similar to T26A, the S36A and S68A single mutants could not activate MST2 in the presence of STK25 (*Figure 4—figure supplement 1A*). The T26A/S68A double mutant only slightly increased pT180 of MST2 (*Figure 4—figure supplement 1B*). We thus mutated all three phosphorylation sites, namely T26, S36, and S68, to alanine (SAV1 3A). SAV1 3A increased pT180 of MST2 more efficiently than SAV1 WT did, and this stimulation was no longer inhibited by STK25 overexpression (*Figure 4A* and *Figure 4—figure supplement 1B*).

We also mutated the same three sites to glutamate (SAV1 3E) to mimic phosphorylation. The SAV1 3E mutant was defective in stimulating pT180 of MST2 (*Figure 4B*). SAV1 3E was also deficient in binding to PP2AA and PP2AC subunits (*Figure 4C*). Finally, recombinant SAV1 PID 3E (pSAV1$^{1-90}$ $^{(3E)}$) partially co-fractionate with PP2AA during gel filtration chromatography (*Figure 3H* and *Figure 3—figure supplement 1E*). These results suggest that phosphorylation of SAV1 by STK25 at multiple sites blocks the association between SAV1 and PP2A, thereby counteracting the ability of SAV1 to inhibit PP2A.

## Discussion

A key initiating event of Hippo signaling is the activation of MST1/2. We have previously shown that the STRIPAK$^{SLMAP}$ complex binds the phosphorylated MST2 linker and reverses MST1/2 activation in a feedback mechanism (*Bae et al., 2017*). The Hippo pathway component SAV1 promotes MST1/2 activation by antagonizing STRIPAK. It is unclear how the PP2A-inhibitory activity of SAV1 is restrained in proliferating cells. In this study, we show that, as a component of STRIPAK, the GCKIII kinase STK25 phosphorylates SAV1 on multiple sites and neutralizes the ability of SAV1 to inhibit STRIPAK (*Figure 4D*). Inactivation of STK25 leads to unrestrained SAV1 activity and spontaneous MST1/2 activation, even without contact inhibition or other stimuli.

SAV1 may not be the only substrate of STK25 in the Hippo pathway. It remains to be tested whether STK25 can phosphorylate other STRIPAK components and if so, whether these phosphorylation events also regulate the conformation and activity of STRIPAK. Furthermore, STK25 may also

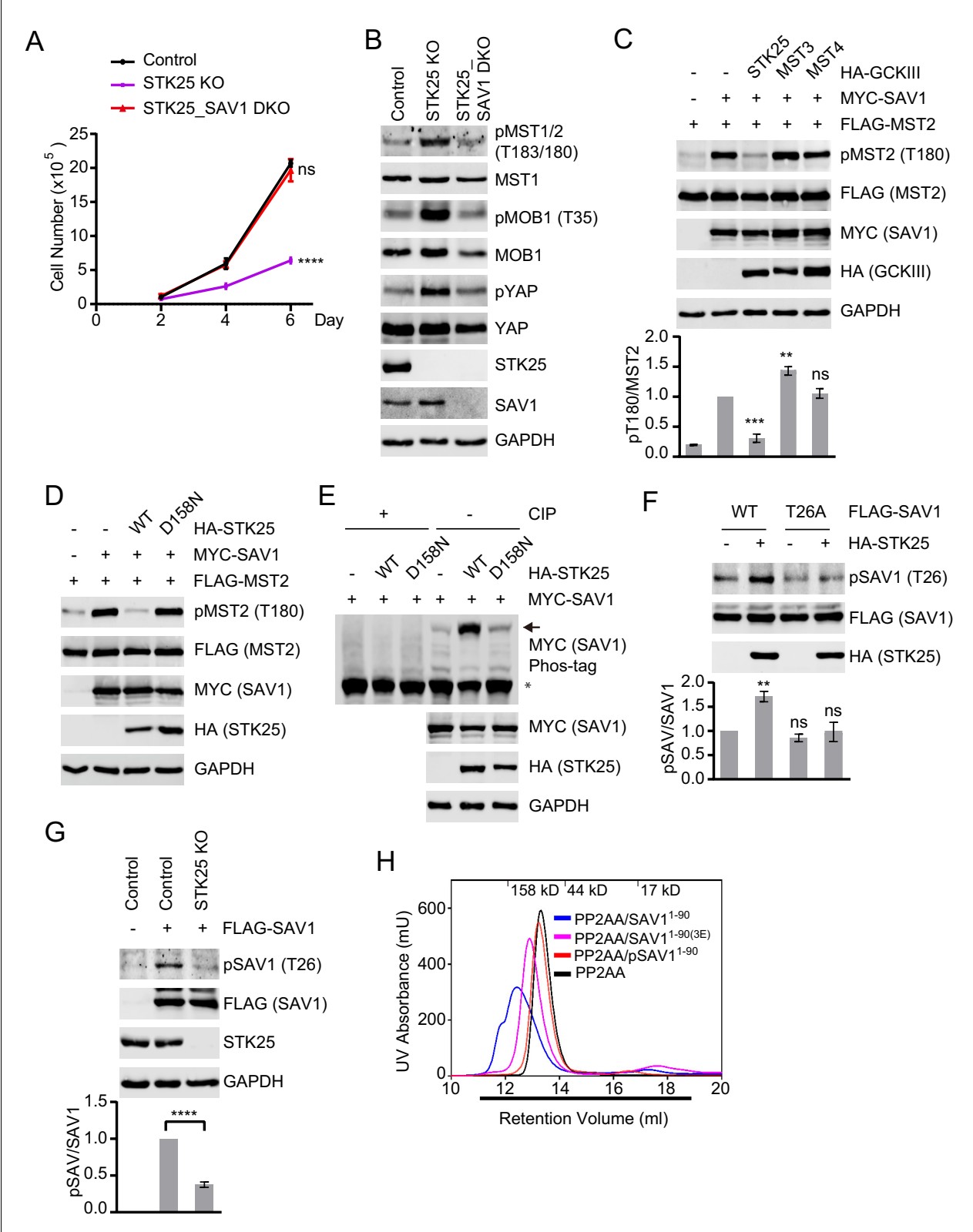

**Figure 3.** STK25 inhibits the ability of SAV1 to counteract STRIPAK. (**A**) Cell proliferation assay. Cell proliferation curves in control (black), STK25 KO (purple), and STK25_SAV1 DKO (red) 293A cells were plotted, respectively. Cells were counted on days 2, 4, and 6 after seeding. Data shown are the means ± SEM of three independent experiments. Numbers of STK25 KO or STK25_SAV1 DKO cells on day 6 was compared to that of control cells (****p<0.0001; ns, non-significant). (**B**) Immunoblots of control, STK25 KO, and STK25_SAV1 DKO 293A cell lysates with the indicated antibodies. (**C**)
*Figure 3 continued on next page*

*Figure 3 continued*

Immunoblots of lysates of 293FT cells co-transfected with FLAG-MST2 and the indicated plasmids. The relative anti-pT180 intensities were normalized to MST2. Data are plotted as mean ± SEM of three biological replicates (***p<0.001; **p<0.01; ns, non-significant). (D) 293FT cells were co-transfected with FLAG-MST2 and the indicated plasmids. The total cell lysates were subjected to immunoblotting. (E) 293FT cells were co-transfected with MYC-SAV1 and HA-STK25. The cell lysates were treated with or without CIP (Calf Intestinal Alkaline phosphatase). CIP-treated and non-CIP-treated lysates were subjected to phos-tag gel analysis and immunoblotting. Asterisk and arrow designate non-phosphorylated SAV1 and phosphorylated SAV1, respectively. (F) Immunoblots and quantification of phosphorylated SAV1 (T26) levels of 293FT cells co-transfected with the indicated plasmids. (G) Control and STK25 KO 293A cells were transfected with FLAG-SAV1. The total cell lysates were blotted with the indicated antibodies. (F and G) The relative anti-pSAV1 intensities were normalized to SAV1. Data are plotted as mean ± SEM of three biological replicates (****p<0.0001; **p<0.01; ns, non-significant). (H) UV traces of the PP2AA and SAV1$^{1-90}$ complex (blue line), PP2AA and phosphorylated-SAV1$^{1-90}$ mixture (pSAV1$^{1-90}$, red line), PP2AA and SAV1$^{1-90(3E)}$ mixture (magenta line), and PP2AA alone (black line) fractionated on a Superdex 200 gel filtration column, respectively. Corresponding molecular weight standards are indicated in kDa. The running buffer for Superdex 200 column containing 20 mM Tris (pH 8.0), 30 mM NaCl, and 1 mM TCEP. The underlined fractions were separated on SDS-PAGE and stained with Coomassie shown in *Figure 3—figure supplement 1E*.

The online version of this article includes the following figure supplement(s) for figure 3:

**Figure supplement 1.** STK25 inhibits the ability of SAV1 to counteract STRIPAK.

**Figure supplement 2.** SAV1 partially co-localizes with STK25 at the Golgi apparatus.

play an indirect role in Hippo regulation. Many components of the STRIPAK complex have been shown to localize to the Golgi (*Baillat et al., 2001*; *Preisinger et al., 2004*; *Fidalgo et al., 2010*; *Matsuki et al., 2010*; *Frost et al., 2012*). Inactivation or depletion of STK25 has been reported to disassemble the Golgi apparatus (*Fidalgo et al., 2010*; *Matsuki et al., 2010*), which may in turn lead to the disassembly and inactivation of the STRIPAK complex. Importantly, phosphorylation of several STRIPAK components is increased in the presence of PP2A inhibitors (*Moreno et al., 2001*), suggesting that the phosphorylation status of the STRIPAK complex might be dynamically regulated to modulate Hippo signaling output.

A recent report by *Lim et al. (2019)* has argued that STK25 activates LATS1/2 by phosphorylating their activation loop (AL) without the phosphorylation of their hydrophobic motif (HM). This proposed mechanism is inconsistent with the well-established two-step sequential activation scheme of NDR kinases, including LATS1/2. In this scheme, upstream kinases, such as MST1/2, first phosphorylate the highly conserved HM of LATS1/2, and the phosphorylated HM then promotes the autophosphorylation of LATS1/2 AL and full activation of LATS1/2. Both HM and AL phosphorylation events are essential for LATS1/2 activation (*Bichsel et al., 2004*; *Chan et al., 2005*; *Stegert et al., 2005*; *Hergovich et al., 2006*; *Hergovich and Hemmings, 2009*; *Ni et al., 2015*; *Hoa et al., 2016*). Our results also contradict the findings of Lim et al. We show that STK25 suppresses Hippo pathway activation at the level of MST1/2, instead of activating the pathway at the level of LATS1/2. The reasons underlying this discrepancy are unclear at present. Future studies are needed to reconcile these disparate findings.

Interestingly, only STK25, but not the highly homologous MST3 and MST4, regulates Hippo pathway activation, although each GCKIII kinase is known to be incorporated into the STRIPAK complex (*Kück et al., 2019*). Since we do not know the relative expression levels of the GCKIII kinases in cells, STK25 may simply be expressed at a much higher level compared with MST3 and MST4, making STK25 the dominant kinase in STRIPAK. Several previous studies have also reported distinctive functions for the GCKIII kinases (*Preisinger et al., 2004*; *Zhou et al., 2009b*; *Fidalgo et al., 2012*; *Mardakheh et al., 2016*). Another possible explanation is that different subcellular localization patterns of the GCKIII kinases produce different functions. For example, MST3 is evenly distributed through the cytoplasm (*Preisinger et al., 2004*). Serum starvation relocates MST4 from the cytoplasm to the Golgi (*Mardakheh et al., 2016*). STK25 localizes to the Golgi apparatus where it partially co-localizes with SAV1 (*Figure 3—figure supplement 2B and C*). This Golgi co-localization may offer a venue for STK25 to specifically phosphorylate SAV1. In support of this idea, 14-3-3ζ localized at the Golgi apparatus is specifically phosphorylated by STK25, but not MST4 (*Preisinger et al., 2004*). Therefore, the GCKIII kinases appear to perform different cellular functions at distinct cellular locations. We also noticed that SAV1 and STK25 are clustered at the peri-nuclear region (*Figure 3—figure supplement 2C*). Components of the STRIPAK complex, including STK25, are known to localize to both the Golgi apparatus and the peri-nuclear region (*Preisinger et al.,*

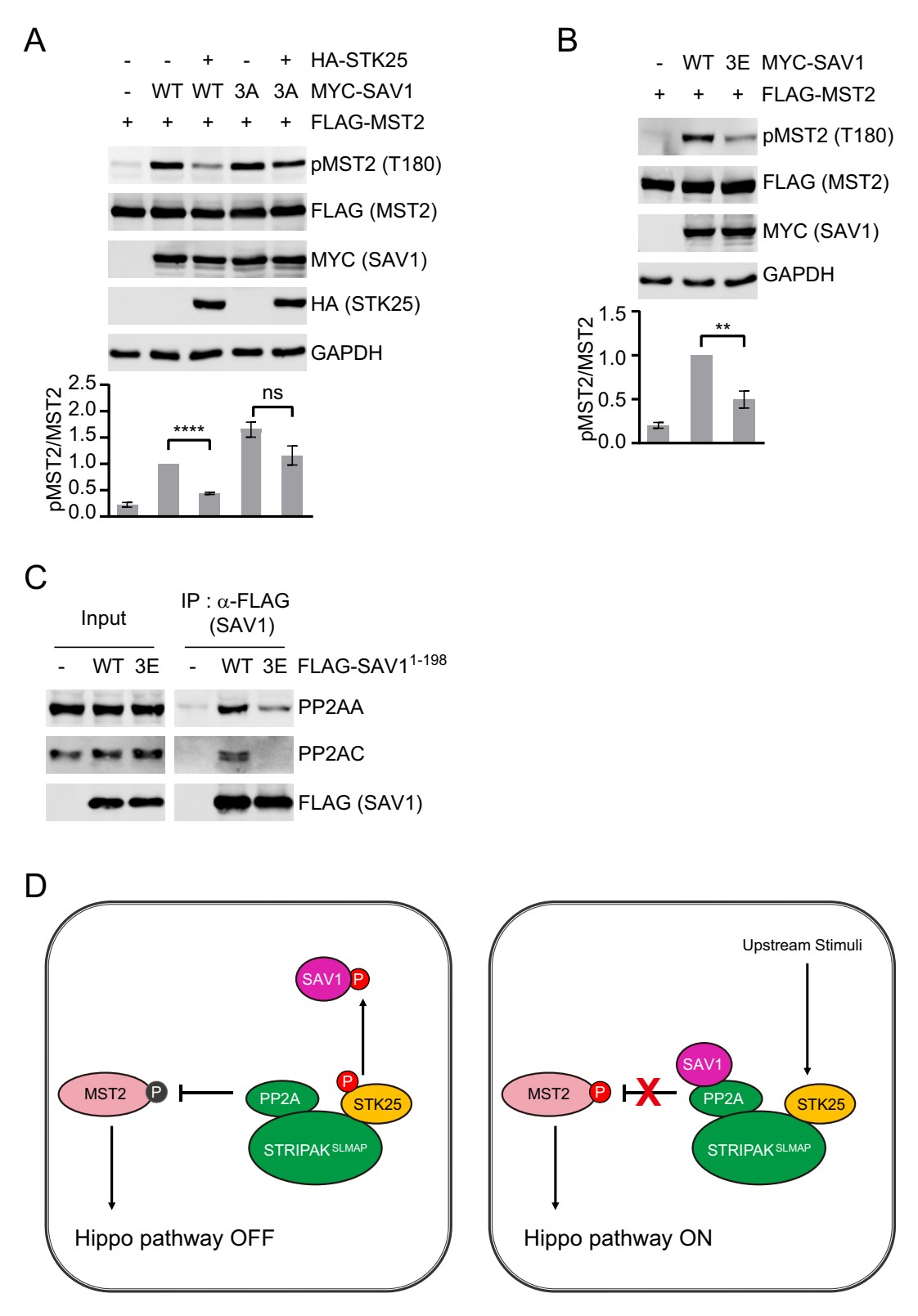

**Figure 4.** SAV1 phosphorylation by STK25 inhibits SAV1-dependent activation of MST1/2. (**A and B**) Immunoblots and quantification of pT180 levels of lysates of 293FT cells co-transfected with FLAG-MST2 and the indicated plasmids. The relative anti-pT180 intensities were normalized to MST2. Data are plotted as mean ± SEM of three biological replicates (****p<0.0001; **p<0.01; ns, non-significant). (**C**) 293FT cells were transfected with the

*Figure 4 continued on next page*

*Figure 4 continued*

indicated FLAG-SAV1 plasmids. The total cell lysates (input) and anti-FLAG IP were blotted with the indicated antibodies. (D) Model for regulation of SAV1-STRIPAK antagonism by STK25.

The online version of this article includes the following figure supplement(s) for figure 4:

**Figure supplement 1.** Single or double mutants of SAV1 are inhibited by STK25.
**Figure supplement 2.** CRISPR/Cas9-induced indel mutations in 293A cells.

*2004*; *Frost et al., 2012*; *Kück et al., 2019*). These regions likely provide specific sites for STK25 to assemble into the STRIPAK complex to regulate Hippo activation.

Because diverse physiological conditions can impinge on the activity of the Hippo pathway, STK25-mediated inhibition of MST1/2 may be regulated by upstream stimuli. For example, oxidative stress has been reported to release STK25 from the Golgi, which is thought to stimulate apoptosis, possibly involving the activation of MST1/2 (*Lehtinen et al., 2006*; *Zhou et al., 2009b*). Many components of the STIRPAK complex localize to the Golgi. Oxidative stress may disrupt the association of STK25 with the STRIPAK complex. This in turn may allow SAV1 to regain its ability to inhibit PP2A and activate MST1/2 to promote apoptosis. Identification of upstream signals of STK25 regulation will deepen our understanding of the Hippo pathway in physiology and disease.

## Materials and methods

### Protein expression and purification

The coding region of human STK25 was cloned into the pET28 vector that included an N-terminal $His_6$ tag. The plasmid was transformed into the bacterial strain BL21 (DE3)-T1$^R$ cells (Sigma). STK25 was purified with Ni-NTA beads (Qiagen) in lysis buffer containing 20 mM Tris (pH 8.0), 300 mM NaCl, 5 mM $MgCl_2$, 10 mM imidazole, and 4 mM 2-mercaptoethanol. The STK25-bound Ni-NTA beads were washed with washing buffer containing 20 mM Tris (pH 8.0), 300 mM NaCl, 5 mM $MgCl_2$, 25 mM imidazole, and 4 mM 2-mercaptoethanol, then eluted with elution buffer containing 20 mM Tris (pH 8.0), 300 mM NaCl, 5 mM $MgCl_2$, 250 mM imidazole, and 4 mM 2-mercaptoethanol. STK25 was further purified using an anion-exchange chromatography with the Resource Q column (GE healthcare). All fractions with purity >90% were collected. Autophosphorylation of STK25 was performed by adding fresh ATP to 2 mM then incubating at room temperature for 30 min. STK25 was further purified with a Superdex 200 size-exclusion column (GE Healthcare) in a buffer containing 20 mM Tris (pH 8.0), 200 mM NaCl, 5 mM $MgCl_2$, and 1 mM TCEP.

### Mass spectrometry

SAV1 PID (SAV1$^{1-90}$) was purified as previously described (*Bae et al., 2017*). SAV1 PID (10 µM) was phosphorylated by recombinant human STK25 (1 µM) at room temperature for 30 min in kinase reaction buffer containing 50 mM Tris (pH 7.5), 150 mM NaCl, 10 mM $MgCl_2$, 1 mM ATP, and 1 mM DTT. The reaction mixture (10 µL) was separated by SDS-PAGE. The band corresponding to phosphorylated SAV1 PID was cut into 1 mm$^3$ cubes. The gel band sample was digested overnight with trypsin (Promega) followed by reduction and alkylation with DTT and iodoacetamine (Sigma-Aldrich). The sample then underwent solid-phase extraction cleanup with an Oasis HLB plate (Waters), and the resulting sample was injected onto an Orbitrap Fusion Lumos mass spectrometer (Thermo Electron) coupled to an Ultimate 3000 RSLC-Nano liquid chromatography system (Dionex). Raw MS data file was analyzed using Proteome Discoverer v 2.2 (Thermo), with peptide identification performed using Sequest HT searching against the human protein database from UniProt. Localization of phosphorylation sites was performed using the PhosphoRS node within Proteome Discoverer. Combined with all peptide information, SAV1 PID (SAV1$^{1-90}$) was fully covered from residue 5 to 80. All sequences with high confidence are listed below, with identified phosphorylation sites specified in parentheses. Based on sequence conservation and MS data, T26, S36, and S68 are chosen as candidates for further analysis.

RTDICLPDSSPNAFSTSGDVVSR (S22); SQCLSTLVRPVFGELK (S5); TDICLPDSSPNAFSTSGDVVSR (S8 and S9); RTDICLPDSSPNAFSTSGDVVSRNQSFLR (S22); NLMPSFIRHGPTIPRRTDICLPDSSPNAF STSGDVVSR (S5); NLMPSFIR (S5); NSQKVTRTLMITYVPK (Y13); HGPTIPRRTDICLPDSSPNAFSTSGD

VVSR (T4); RNTFVGTPFWMAPEVIK (T3); YVKKETSPLLR (T6); GTALHSSQKPAEPVK (S6 and S7); GTA LHSSQKPAEPVK (S6); GTALHSSQKPAEPVKR (T2 and S7); GTALHSSQKPAEPVKR (T2); NEVSK-PAEVQGK (S4); KTKNEVSKPAEVQGK (T2); TKNEVSKPAEVQGK (T1); KKTKNEVSKPAEVQGK (S8); RTDICLPDSSPNAFSTSGDVVSR (S9 and S10); KETSPLLR (T3); HKQSGGSVGALEELENAFSLAEESCPGI SDK (S7); NLMPSFIR (S5).

## In vitro kinase assays

LATS1C-ΔHM (2 µM) was treated by MST2 (0.5 µM), STK25 (0.5 µM) or pLATS1-HM (4 µM), respectively, in kinase reaction buffer containing 50 Tris (pH 7.5), 150 mM NaCl, 10 mM MgCl$_2$, 1 mM ATP, and 1 mM DTT. Reaction mixtures (5 µL) were separated by SDS-PAGE and blotted with an antibody against pS909 of LATS1. SAV1$^{1-90}$ or SAV1$^{1-90}$ T26A (10 µM) was phosphorylated by STK25 (1 µM) at room temperature for 30 min in kinase reaction buffer containing 50 mM Tris (pH 7.5), 150 mM NaCl, 10 mM MgCl$_2$, 1 mM ATP, and 1 mM DTT, respectively. The reaction mixture (3 µL) was separated by SDS-PAGE and blotted with an antibody against pT26 of SAV1. The membrane was then stripped and re-blotted with the His-tag antibody as the loading control.

## Mammalian cell culture and transfection

293FT cells (R70007, Thermo Scientific; not independently authenticated) and 293A cells (R70507, Thermo Scientific; not independently authenticated) were cultured at 37˚C in a humidified 5% CO$_2$ atmosphere. All cells were maintained in DMEM supplemented with 10% fetal bovine serum, 2 mM L-glutamine, and 1% penicillin/streptomycin. All cell lines were checked by Hoechst 33342 (H3570, Thermo Scientific) staining to ascertain that they were free of mycoplasma contamination. Lipojet (Signagen) was used for transient transfection according to the manufacturer's instructions. Cells were harvested for further experiments after 24 hr.

For Hippo signaling analysis, $5 \times 10^5$ cells were plated on 60 mm dishes. After 24 hr, cells were collected for immunoblotting and qRT-PCR after we had confirmed that cells were cultured as sparse sub-confluent monolayers (*Figures 1A, D, F* and *3B*). For activation of the Hippo pathway by cytoskeleton disruption, DMSO, latrunculin B (1 µg/ml), or cytochalasin D (0.4 µM) was added for 1 hr before cells were collected (*Figure 1—figure supplement 3A*). $1 \times 10^5$ cells were seeded on Lab-Tek II four well chamber slides for immunofluorescence. After 24 hr, cells were fixed for further experiments.

siRNAs siRNAs were transfected into 293FT cells with Lipofectamine RNAiMAX (Invitrogen) according to manufacturer's instructions. 48 hr later, FLAG-MST2 was transfected with the Lipoject reagent. Cells were collected for immunoblotting at 24 hr after transfection. The following siRNAs were synthesized by Dharmacon and used in this study: si*STK25* #1, GAAGGUGCCCUGUGCUAUG dTdT; si*STK25*#2, CGGAGCAGGGUGACGUGAAdTdT; si*MST3* #1, UCCCAAGAAUCUCGAGAA UdTdT; si*MST3* #2, UGAAUAAGGAGCCGAGCUUdTdT; si*MST4* #1, GAUUGAAGAACUCGAGAAA dTdT; si*MST4* #2, CACCAAACCUACGUCAAGAdTdT; si*SIKE1*, CCUGAAAGCUCACCAGUCUdTdT; si*CCM3*, UAUGGCAGCUGAUGAUGUAdTdT.

## Generation of knockout cell lines

The sgRNAs of STK25, MST3, and MST4 were cloned into the psCas9 vector (Addgene). The individual psCas9-sgRNA plasmids were transfected into 293A cells with the Lipojet reagent. One day after transfection, cells were selected with 1 µg/ml puromycin. After two days of selection, single cells were re-plated into individual wells of 96-well plates without puromycin. Single clones were tested by immunoblotting and DNA sequencing. For generating the STK25_MST1/2 and STK25_SAV1 knockout cell lines, the sgRNAs of MST1, MST2, and SAV1 were cloned into the plentiCRISPR v2 vector (Addgene). The plentiCRISPR v2-sgRNA, pMD2.G, and psPax2 plasmids were co-transfected into 293FT cells with the Lipofectamine 2000 reagent (Invitrogen). Two or three days after transfection, the lentiviral supernatants were harvested and concentrated with the Lenti X-concentrator (Clontech). STK25 knockout cells were infected with the lentiviruses and 4 µg/ml polybrene. Two days after infection, cells were selected with 1 µg/ml puromycin. After three days of selection, single cells were sorted into individual wells of 96-well plates. Single clones were tested by immunoblotting and DNA sequencing (*Figure 4—figure supplement 2*).

The following sgRNAs were used in this study: STK25, CGAATCGGGGGTCTTTGTTG; MST3, AATACTTACTAGATCTAGTG; MST4, GGATGCATATCGGAGTTAGG; MST1, ATACACCGAGATATCAAGGC; MST2, AGTACTCCATAACAATCCAG; SAV1, GGAGGTGGTTGATCATACCG.

## Antibodies and reagents

Rabbit polyclonal MST2 phospho-T336 antibody was previously described (*Bae et al., 2017*). Rabbit polyclonal SAV1 phospho-T26 was raised against the SAV1 phospho-peptide with the sequence of VKKEpTSPLLC at an on-campus facility. The following antibodies were purchased from the indicated sources: anti-pMST1/2 (T183/180; GTX133948, GeneTex); anti-STRIP1 (A304-644A) and anti-CCM3 (A304-798A, Bethyl Laboratories Inc); anti-Tubulin (ab4074), anti-SIKE1 (ab121860), anti-STK25 (ab157188) and anti-SLMAP (ab56328, Abcam); anti-MYC (Roche); anti-FLAG (F1804, Sigma); anti-HA (sc-805), anti-PP2AA (sc-6112), anti-SAV1 (sc-101205), anti-STRN3 (sc-13562) and anti-YAP (sc-101199, Santa Cruz Biotechnology); anti-MOB4 (A4590, ABclonal); anti-GM130 (12480), anti-MST1 (3682), anti-MST2 (3952), anti-MST3 (3723), anti-MST4 (3822), anti-GAPDH (2118), anti-MOB1 (13730), anti-pMOB1 (T35;8699), anti-LATS1 (9153), anti-pLATS1/2 (HM; 8654, AL;9157), anti-pYAP (4911), anti-PP2AC (2259), anti-rabbit immunoglobulin G (IgG) (H+L) (Dylight 800 or 680 conjugates), and anti-mouse IgG (H+L) (Dylight 800 or 680 conjugates, Cell Signeling). Alexa Fluor 647 Phalloidin (A22287) was purchased from Thermo Scientific. Phos-tag conjugated acrylamide was purchased from Wako chemicals. Latrunculin B (ab144291) and cytochalasin D (sc-201442) were purchased from Abcam and Santa Cruz Biotechnology, respectively.

## Immunoblotting and immunoprecipitation

For immunoblotting, cell lysates and immunoprecipitates were analyzed by standard immunoblotting protocol. The membranes were scanned and band intensities were quantified by an Odyssey Infrared Imaging System (LI-COR). For immunoprecipitation, cells were harvested and lysed with the lysis buffer containing 20 mM Tris-HCl (pH 7.5), 150 mM NaCl, 0.2% Triton X-100, protease inhibitors (Roche), and PhosSTOP (Roche) on ice for 20 min. After incubation, cell lysates were separated by centrifugation at 20,000 g for 20 min at 4°C. Cleared cell lysates were incubated with anti-FLAG M2 resin (Sigma) or anti-STRN3-crosslinked protein A resin for 2 hr at 4°C. After incubation, resins were washed by the washing buffer containing 20 mM Tris-HCl (pH 7.5), 150 mM NaCl, 0.1 mM EDTA, and 1% Triton X-100 and eluted by SDS sampling buffer. Eluates were separated by SDS-PAGE and blotted with the appropriate antibodies.

## Immunofluorescence

Cells were washed once with PBS and fixed with 4% paraformaldehyde for 20 min. Cells were then permeabilized with PBS containing 0.2% Triton X-100 (PBS-T) for 20 min and blocked with PBS-T containing 3% BSA for 30 min at room temperature. Cells were incubated with the primary antibody in PBS-T containing 3% BSA for overnight at 4°C. Cells were then washed with PBS-T three times and incubated with the secondary antibody for 1 hr. Cells were washed again with PBS-T three times and mounted in the ProLong Gold Antifade reagent with DAPI (Invitrogen). Cells were visualized with a DeltaVision microscope system (Applied Precision). For assaying YAP localization, approximately 100 cells were counted in randomly chosen fields for each sample. The fluorescence intensity of YAP was measured in the cytoplasm and nucleus using Image J, and the subcellular localization of YAP was determined based on the measured intensities. Experiments were repeated three times for statistical analysis. For F-actin staining, cells were incubated with Alexa Fluor 647-conjugated phalloidin for 1 hr after blocking with BSA. Cells were washed with PBS two times and mounted. Cell area was measured using Image J based on F-actin staining.

## Real-time qRT-PCR

Total RNA was isolated from cells using the Trizol reagent (Invitrogen). cDNA was obtained by reverse transcription reactions using the reverse transcription kit (Applied Biosystems). Real-time PCR was performed using iTaq Universal SYBR Green Supermix (Bio-Rad) and the 7900HT Fast Real-Time PCR System (Applied Biosystems). The relative abundance of mRNA was normalized to GAPDH.

## Dephosphorylation assay

1000 U of calf intestinal phosphatase (CIP; M0290, New England Biolabs) was mixed with 100 mg of cell lysates in CIP buffer [50 mM Tris-HCl (pH 7.9),100 mM NaCl, 10 mM MgCl$_2$, 1 mM Dithiothreitol (DTT)] for 12 hr at 37°C. After incubation, SDS sample buffer was added into the mixture to stop the reaction. Samples were boiled for 5 min at 37°C and subjected to phos-tag gel analysis.

## Cell proliferation assay

293A cells were seeded at a density of $2.5 \times 10^4$ cells per well on 6-well dishes. Cells were cultured in DMEM supplemented with 10% fetal bovine serum, 2 mM L-glutamine and 1% penicillin/strepto-mycin. Cells were counted on days 2, 4, and 6 after seeding using TC20 automated cell counter (BioRad).

## Statistical analysis

Values are presented as mean ± SEM from at least three biological replicates. Results were evaluated by two-tailed unpaired t tests. The graphs and statistical calculations were performed using Prism (GraphPad).

## Acknowledgements

We thank Hongtao Yu for critical reading of the manuscript. We are grateful to Yonggang Zheng and Duojia Pan for helpful discussion. This work was supported in part by grants from the National Institutes of Health (GM107415 and GM132275 to XL), and the Welch Foundation (I-1932 to XL)

## Additional information

### Funding

| Funder | Grant reference number | Author |
| --- | --- | --- |
| National Institute of General Medical Sciences | GM107415 | Xuelian Luo |
| National Institute of General Medical Sciences | GM132275 | Xuelian Luo |
| Welch Foundation | I-1932 | Xuelian Luo |

The funders had no role in study design, data collection and interpretation, or the decision to submit the work for publication.

### Author contributions

Sung Jun Bae, Conceptualization, Data curation, Formal analysis, Validation, Investigation, Visualization, Methodology, Writing - original draft; Lisheng Ni, Data curation, Formal analysis, Validation, Investigation, Visualization, Methodology; Xuelian Luo, Conceptualization, Resources, Formal analysis, Supervision, Funding acquisition, Validation, Investigation, Project administration, Writing - review and editing

### Author ORCIDs

Xuelian Luo (iD) https://orcid.org/0000-0002-5058-4695

### Decision letter and Author response

Decision letter https://doi.org/10.7554/eLife.54863.sa1
Author response https://doi.org/10.7554/eLife.54863.sa2

## Additional files

### Supplementary files

• Transparent reporting form

## Data availability

All data generated or analyses during this study are included in the manuscript and supporting files.

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
