## [Decision Letter]

**Acceptance summary:**

The authors have responded carefully to all points raised in the original review, including adding several new experiments. The data are convincing that the STK25 kinase component of the PP2A STRIPAK complex down regulates the Hippo pathway by phosphorylating Sav1, which prevents Sav1 from inhibiting PP2A dephosphorylation of Mst_1/2_, which down regulates Mst_1/2_ activity (the results shown in their Bai et al., 2017 *eLife* paper). They rule out other effects on STRIPAK on the actin cytoskeleton. This work adds another layer to the complex feedback that regulates the Hippo pathway.

**Decision letter after peer review:**

Thank you for submitting your article "YSK1 suppresses Hippo signaling by regulating SAV1-STRIPAK antagonism" for consideration by *eLife*. Your article has been reviewed by three peer reviewers, one of whom is a member of our Board of Reviewing Editors, and the evaluation has been overseen by Philip Cole as the Senior Editor. The reviewers have opted to remain anonymous.

The reviewers have discussed the reviews with one another and the Reviewing Editor has drafted this decision to help you prepare a revised submission.

Summary:

This is a significant advance from the Bae et al., 2017 paper. There these authors showed that Sav1 helps to activate the Mst_1/2_ kinase by inhibiting its dephosphorylation by the STRIPAK PP2A phosphatase complex. Here, they show that the Ysk1 kinase component of the PP2A STRIPAK complex down regulates the Hippo pathway. They show that Ysk1 phosphorylates Sav1, which prevents Sav1 from inhibiting PP2A dephosphorylation of Mst_1/2_, thereby down regulating Mst_1/2_ activity. This provides a dynamic feedback mechanism to regulate the pathway.

Essential revisions:

1) Since the published literature has mostly focused on STK25 as influencing Golgi morphology and participating in oxidative stress response, could some of the reported effects on Hippo actually be indirect consequences of events occurring downstream of STK25? It would be particularly helpful to know whether STK25 has any effects on the F-actin cytoskeleton, given its central role in Hippo pathway regulation.

2) The experiments monitoring Hippo signaling don't include assessments of cell density (at least not adequately described in the Materials and methods or figure legends). When making comparisons between different experimental conditions, we need to know whether the cell density is the same, as variations in cell density can have substantial effects on Hippo signaling.

3) Based on knockouts in 293FT cells. they conclude that of the three GCKIII kinases, YSK1 is the only one that affects Sav1 activity. There is no information provided as to the relative expression levels of these three kinases in these cells, nor whether their relative expression is similar in other cell types, or whether each can incorporate in the STRIPAK complex. It should be clarified if this is already known in the literature. If not, the authors should be more circumspect in their claims. The brief mention of subcellular localization differences in the Discussion doesn't address this fundamental issue.

4) The experiments in Figure 4, which are key to the inhibitory mechanism, are incomplete. Using in vitro phosphorylation with recombinant kinase, they identify several phosphorylation sites and raise an antibody to a peptide containing one of these phosphorylated sites (T26). However, the T26A mutant has no effect, so they mutate Thr26 along with two other sites identified by mass spec, and subsequently observe functional effects. They imply that all of these sites might contribute weakly to the functional effects, but since they are rather dispersed in the PID sequence, it seems quite plausible that one of the non Thr26 phosphorylation sites is key. It would be helpful for the authors to state why they seem not to have mutated the other two sites individually. Such information would solidify the mechanism. They should also discuss the relative conservation of the sites in the PID. Also, how complete was the coverage in the mass spec analysis? Could there be other sites?

5) The Materials and methods to follow and/or reproduce the work are insufficient.

a) A large portion of their story is the identification of the phosphorylation sites on hSav by YSK1. However a detailed description of the methods used for generating the data (concentrations of enzyme/substrate a and details of the MS methods) is lacking and the MS data not provided.

b) The authors do not provide information on IF staining and the quantification (Figure 1B-C) – e.g. are they simply counting raw pixel values? How are images are processed etc.?

c) Materials and methods or standardization for "RNA expression" (Figure 1D, G), IP assays (Figure 2), for over-expression assays (Figure 1—figure supplement 1), and complex formation and gel filtration (Figure 3H) were not found. It would be difficult to for other investigators to reproduce the data presented.

6) Although it doesn't affect the qualitative conclusions, many of the experiments claiming significant differences appear to use a p value threshold of 0.1. The authors may wish to be more cautious in concluding effects based on such a weak threshold.

---

## [Author Response]

Essential revisions:1) Since the published literature has mostly focused on STK25 as influencing Golgi morphology and participating in oxidative stress response, could some of the reported effects on Hippo actually be indirect consequences of events occurring downstream of STK25? It would be particularly helpful to know whether STK25 has any effects on the F-actin cytoskeleton, given its central role in Hippo pathway regulation.

We thank the reviewers for this very insightful comment. We tested the effect of STK25 knockout on the F-actin cytoskeleton by immunofluorescence. However, we did not observe any significant change of the F-actin cytoskeleton structure between control and STK25 KO cells. These data were presented in the revised Figure 1—figure supplement 4. We have also added the detailed description of the experiments in the revised Materials and methods. Even though we did not observe an effect of STK25 depletion on the actin cytoskeleton, we cannot rule out the possibility of other indirect effects. We have thus revised the text to discuss the possibility that, in addition to SAV1 regulation, STK25 depletion might also affect the Hippo pathway through indirect effects.

2) The experiments monitoring Hippo signaling don't include assessments of cell density (at least not adequately described in the Materials and methods or figure legends). When making comparisons between different experimental conditions, we need to know whether the cell density is the same, as variations in cell density can have substantial effects on Hippo signaling.

We thank the reviewers for this helpful comment. We agree with the reviewers that cell density is critical for Hippo signaling. For our Hippo signaling analysis, we always took great care to ensure that cells were cultured as sparse sub-confluent monolayers before collecting them for further experiments. We have revised the Materials and methods accordingly.

3) Based on knockouts in 293FT cells. they conclude that of the three GCKIII kinases, YSK1 is the only one that affects Sav1 activity. There is no information provided as to the relative expression levels of these three kinases in these cells, nor whether their relative expression is similar in other cell types, or whether each can incorporate in the STRIPAK complex. It should be clarified if this is already known in the literature. If not, the authors should be more circumspect in their claims. The brief mention of subcellular localization differences in the Discussion doesn't address this fundamental issue.

We thank the reviewers for this insightful comment. We found that each GCKIII kinase was detectable by individual antibodies in 293 cells. Unfortunately, an antibody that recognizes all three GCKIII kinases is not available so that we cannot easily determine the relative protein levels of each GCKIII kinase. We have thus modified the text to discuss the possibility that STK25 might be the most abundant member. MST3 and/or MST4 may also be involved in regulation of the Hippo pathway in the context of different cellular localization or cell types. We also added the relevant references on the fact that each GCKIII kinase can be incorporated into the STRIPAK complex.

4) The experiments in Figure 4, which are key to the inhibitory mechanism, are incomplete. Using in vitro phosphorylation with recombinant kinase, they identify several phosphorylation sites and raise an antibody to a peptide containing one of these phosphorylated sites (T26). However, the T26A mutant has no effect, so they mutate Thr26 along with two other sites identified by mass spec, and subsequently observe functional effects. They imply that all of these sites might contribute weakly to the functional effects, but since they are rather dispersed in the PID sequence, it seems quite plausible that one of the non Thr26 phosphorylation sites is key. It would be helpful for the authors to state why they seem not to have mutated the other two sites individually. Such information would solidify the mechanism. They should also discuss the relative conservation of the sites in the PID. Also, how complete was the coverage in the mass spec analysis? Could there be other sites?

We thank the reviewers for this very insightful comment. We have included SAV1 WT, all three SAV1 individual mutants (T26A, S36A, and S68A), a double mutant (T26A and S68A), and a triple mutant (T26A, S36A, and S68A) in our new experiments to compare their effects on the pT180 level of MST2. These new data were presented in the revised Figure 4—figure supplement 1. The coverage of SAV1 PID in the mass spectrometry analysis was complete. The relevant legend and method were revised accordingly.

5) The Materials and methods to follow and/or reproduce the work are insufficient.a) A large portion of their story is the identification of the phosphorylation sites on hSav by YSK1. However a detailed description of the methods used for generating the data (concentrations of enzyme/substrate a and details of the MS methods) is lacking and the MS data not provided.

We thank the reviewers for this helpful comment. We have added a detailed description of our MS methods in the revised Materials and methods. We could not provide the MS/MS spectrum for the pT26 peptide since it is no longer available.

b) The authors do not provide information on IF staining and the quantification (Figure 1B-C) – e.g. are they simply counting raw pixel values? how are images are processed? etc.

We thank the reviewers for this helpful comment. We have revised the Materials and methods to provide additional information on immunofluorescence.

c) Methods or standardization for "RNA expression" (Figure 1D, G), IP assays (Figure 2), for over-expression assays (Figure 1—figure supplement 1), and complex formation and gel filtration (Figure 3H) were not found. It would be difficult to for other investigators to reproduce the data presented.

We thank the reviewers for this helpful comment. We have revised the Materials and methods to provide detailed information on these experiments.

6) Although it doesn't affect the qualitative conclusions, many of the experiments claiming significant differences appear to use a p value threshold of 0.1. The authors may wish to be more cautious in concluding effects based on such a weak threshold.

We thank the reviewers for catching our mistake. * actually indicated P<0.05, not P<0.1 as we stated. We have corrected this mistake in the relevant legend and method.